# Effect of Consuming Beef with Varying Fatty Acid Compositions as a Major Source of Protein in Volunteers under a Personalized Nutritional Program

**DOI:** 10.3390/nu14183711

**Published:** 2022-09-09

**Authors:** Diana A. Vela-Vásquez, Ana M. Sifuentes-Rincón, Iván Delgado-Enciso, Cynthia Ordaz-Pichardo, Williams Arellano-Vera, Víctor Treviño-Alvarado

**Affiliations:** 1Animal Biotechnology Laboratory, Center for Genomic Biotechnology, Instituto Politécnico Nacional, Reynosa 88710, Mexico; 2Faculty of Medicine, University of Colima, Colima 28040, Mexico; 3State Institute of Cancerology of the Health Services of the State of Colima, Colima 28085, Mexico; 4National School of Medicine and Homeopathy, Instituto Politécnico Nacional, Mexico City 07320, Mexico; 5School of Medicine, Tecnológico de Monterrey, Monterrey 64710, Mexico; 6The Institute for Obesity Research, Tecnológico de Monterrey, Monterrey 64710, Mexico

**Keywords:** beef, monounsaturated fatty acids, saturated fatty acids, food intervention, weight-loss program

## Abstract

Beef is an excellent source of nutrients; unfortunately, most nutritional recommendations suggest limiting or even avoiding it. Studies have shown that the fatty acid composition of meat influences weight loss. This randomized controlled clinical trial evaluated the anthropometric and serum lipid changes after a food intervention that included frequent beef consumption (120 g consumed four days/week for four weeks). Volunteers were randomly assigned to the commercial or Wagyu-Cross beef groups, with the latter beef possessing higher fat and MUFA contents. Both groups exhibited reductions in body measurements and lipid profiles; however, the Wagyu-Cross group exhibited greater changes in weight (−3.75 vs. −2.90 kg) and BMI (−1.49 vs. −1.03) than the commercial group, without a significant difference between them. No significant group differences in lipid profiles were observed; however, the Wagyu-Cross group exhibited a more favorable change in decreasing the TC concentration (−7.00 mg/dL) and LDL-C concentration (−12.5 mg/dL). We suggest that high MUFA beef could be included in weight-loss programs since it does not affect weight loss and hasn’t a negative influence on lipid metabolism.

## 1. Introduction

Obesity is a complex chronic disease related to genetics, in which individuals exhibit a positive energy intake-to-expenditure ratio, indicating that more energy is consumed than used [1]. Obesity is a growing problem worldwide, and in certain countries, such as Mexico, the obesity rates are alarming; recent reports found that seven out of ten adults are obese, with a prevalence of 35% and 46% for men and women, respectively [2]. Being overweight or obese is associated with an increased risk of developing type 2 diabetes mellitus, high blood pressure, cardiovascular diseases, and premature death [3,4].

Diet is an important factor in obesity development as well as control [5], and caloric restriction is the main approach to achieve weight loss [6]. Reductions in BMI and waist circumference are important to reduce obesity and central obesity, which are modifiable risk factors for non-communicable diseases such as diabetes and hypertension [7]. The evidence has shown that reducing the fat in dairy products and increasing the consumption of high-fiber foods result in smaller increases in body mass index (BMI) in women and waist circumference in both sexes than following a food pattern rich in white bread and refined grains, processed meat, potatoes, and meat [8]. On the other hand, changes are also observed in individuals following a dietary pattern with a low consumption of fat-rich foods (red meat, margarine, butter, etc.) and higher consumption of fiber from cereals, vegetables, bread, etc. [9]; therefore, when designing dietary programs, it is crucial to identify the products that provide better outcomes in terms of weight reduction and patient well-being.

On average, Mexican adults obtain 16% of their energy from products high in saturated fat or added sugar and 14% of their energy from foods of animal origin [10].

Beef is an excellent source of high-quality protein, essential nutrients (iron, zinc, selenium, potassium, and vitamin B), and fatty acids (FAs) [11]; however, limiting or even avoiding beef consumption are constants in weight-loss programs, with the main reasons for rejection being beef’s saturated fatty acid (SFA) concentration and calorie density [12]. From a public health perspective, the high SFA composition of beef has been the main limiting factor between beef consumers because it has been associated with a higher probability of developing coronary heart disease (CHD) or cardiovascular disease (CVD) [13,14], and cancer risk [15]. However, it is important to mention that new evidence in a meta-analysis of cohort studies has found no association, or reduced risk, between the consumption of unprocessed red meat (including beef) and processed meat and mortality and CVD [16], and other environmental factors, such as alimentary and lifestyle habits, emerge as focal points to be considered as risk factors [17].

Recent data suggest that protein sources such as beef are highly effective for decreasing body weight due to satiety and reduced appetite [18,19,20].

Given the importance of the type of dietary fat, the effects of the consumption of different FAs on body weight loss have been investigated, although with contradictory results. Reductions in body fat mass but not in weight were observed in response to fish oil consumption by healthy adults [21]. Additionally, Thorsdottir et al. [22] reported greater weight loss and waist circumference reduction in men who consumed fish (lean or fatty) or DHA/EPA capsules compared with the control group who consumed sunflower oil capsules. On the other hand, reduction in weight was also observed in response to the addition of monounsaturated fatty acids (MUFAs) or SFAs to a low-calorie diet, although there was no significant difference between the treatments [23]. Recently, Vela-Vásquez et al. [24] reported slightly greater weight loss and reductions in body measurements in volunteers who consumed beef with different fatty acid compositions, supporting the idea that MUFA consumption, compared with SFA consumption, provides major benefits in weight-loss programs [25,26]. The Wagyu-Cross beef was used by the authors as a high MUFA content beef (45.44% of C18:1 n-9, with a total MUFA content of 51.40% and 0.43% of C15:0, 24.56% of C16:0, 14.01% of C18:0, with a total SFA content of 42.96%), while commercial beef was considered to be beef with a high SFA content (43.09% of C18:1 n-9, with a total MUFA content of 49.59% and 0.61% of C15:0, 26.27% of C16:0, 14.11% of C18:0 with a total SFA content of 45.23%) [24].

The objective of this work was to evaluate changes in anthropometric measurements (body mass and circumferences), serum lipid parameters, and atherogenic indices in volunteers undergoing a food intervention with personalized diets that include frequent consumption of beef with different fatty acid compositions (commercial beef vs. Wagyu-Cross beef).

## 2. Materials and Methods

### 2.1. Study Approval

The study was a randomized, controlled, double-blinded clinical trial carried out according to the Consolidated Standards of Reporting Trials (CONSORT) guidelines. Ethical approval for the study was obtained from the Bioethics Committee of the Escuela Nacional de Medicina y Homeopatía (CBE/007/2021), and all of the subjects provided written, informed consent. The present clinical trial was registered as NUTRIRES2: RPCEC00000403 in the Cuban Public Registry of Clinical Trials (RPCEC) Database.

### 2.2. Volunteers

Thirty-three volunteers were screened to determine their eligibility for the present trial, and all of them were recruited at the Centro de Biotecnología Genómica—Instituto Politécnico Nacional, located in the city of Reynosa, Tamaulipas, Mexico. The inclusion criteria included men or women aged between 25 and 55, with a body mass index (BMI) ≥18.5 kg/m^2^ and <45 kg/m^2^, who wanted to lose weight and/or to improve healthy eating habits, all of whom self-reported that they were frequent beef consumers (i.e., consumed beef once or twice a week) and were willing to pause their consumption of fish, pork, and beef not provided by the clinical trial as well as other food supplements. The exclusion criteria were individuals with type 1 or 2 diabetes, hypertension, chronic or acute diseases at the time of the interview, alcoholism or drug addiction, those who smoked (>10 cigarettes a day), were pregnant or breastfeeding, or who used medical treatments or dietary supplements as evaluated by a self-reported form. In addition, the volunteers who were considered unlikely to comply with the study protocol were excluded.

### 2.3. Study Design

As mentioned above, the trial was double-blinded, and the volunteers, sample collectors, and researchers were not aware of the group of volunteers. The sample size calculation was based on the results described by Varady et al. [27] (α = 0.05, power = 0.8), which were performed on the basis of expected mean ± SD differences in weight loss between diet groups of 6.5% to determine the targeted final sample size (*n* = 6 in each group). After the survey, four participants were excluded because they did not meet the inclusion criteria. The remaining 29 volunteers were randomly assigned to either the control (*n* = 14; commercial beef) or the treatment (*n* = 15; Wagyu-Cross beef) for food interventions. However, to avoid any biases, the analysis was restricted to those volunteers with a BMI of ≥25 kg/m^2^, resulting in the following distribution, *n* = 8; commercial beef and *n* = 12 Wagyu-Cross beef.

### 2.4. Dietary Assessment

During the food intervention, the volunteers were given a 7-day menu to follow, which described the number of meals and amounts of each food (fruits, vegetables, dairy, cereals, etc.) which included beef consumption (120 g consumed four days/week for four weeks). The menus were adjusted according to the volunteers’ energy requirements in a state of caloric restriction (20% average of deficit caloric from their basal metabolic rate, using the daily calorie calculations for body weight—sedentary by Kashi Clinical Laboratories, Inc., Portland, OR, USA). The menus were changed every week, but the maintenance of caloric consumption was strictly supervised. Considering that weight loss is primarily determined by the calories ingested [28], all of the menus were designed having as a basepoint the calories provided by the serving of each type of beef (commercial beef or Wagyu-Cross beef), the rest of the macronutrients (proteins and carbohydrates) were adapted to the required caloric content of each diet (following the conventional diet percentages recommended for healthy adults), ensuring that the caloric deficit was met in each group (commercial or Wagyu-Cross), as well as the differences in fat composition from other diet intakes (beef-fat-free intakes). The menu included chicken, eggs, turkey ham, sausages, panela or cottage cheese, 2% milk (reduced fat), and plain (light or whole milk) Greek yogurt as the animal protein options and beans, peas, lentils, chickpeas, avocados, nuts, oats, bread, and different kinds of vegetables as the non-animal protein options.

Each serving of ground/fajita beef (120 g) for both of the groups was packed in a plastic bag and cold-chain stored until delivery. At the beginning of the trial, each volunteer received 16 packages of meat (4 per week) and were reminded that it was their responsibility to refrigerate the beef until consumption and prepare the beef following the instructions provided (e.g., cooking without added oil). Two cooking methods, in a frying pan and boiling (meatballs), had lower cooking losses and were therefore recommended since SFA and MUFA concentrations increase due to water loss after cooking [29].

Dietary monitoring was carried out weekly, either in person or via cell phone, to review adherence to the diet, and exercise and moderate or intense physical activities were restricted to avoid generating an additional caloric expenditure.

### 2.5. Beef Processing

We previously reported that MUFAs are higher in Mexican Wagyu-Cross beef than in commercially available ground beef [24]. The same beef sources were included in this study, and the quality control of their composition was verified in all of the batches used during the food intervention by bromatological analysis. The commercial beef samples were obtained from a grocery store. Individual 2-inch T-bone steaks were acquired over two weekends (Friday, Saturday, and Sunday) at two different times (8 am and 8 pm) to ensure the random inclusion of different carcasses and thereby represent commercial beef samples. The Wagyu-Cross beef samples were provided by a Wagyu-Cross producer located in Durango, Mexico, and represented animals slaughtered in 2021. Individual cuts were collected to obtain 26.88 kg of ground meat from both sources.

The grinding of these cuts of meat was carried out randomly, ensuring homogeneity, with Migsa grinding equipment, model HFM-12 (Maquinaria Internacional Gastronómica, S.A. de C.V., CDMX, Mexico). The ground meat samples were packed in properly identified plastic bags and stored in a cold chain (−20 °C). All of the beef portions to complete the clinical trial were weighed, packaged, and delivered to each volunteer one day before they started the food intervention.

### 2.6. Outcome Measures and Follow-Up

Venous blood samples were obtained after a 12-h (overnight) fast, both before and after the food intervention, to determine the plasma lipid profile using the services of a particular clinical laboratory where the commercial blood chemistry panel SMAC24 was performed.

The primary endpoint was the change from baseline in the anthropometric measurements and indices: body weight, BMI, waist circumference, hip circumference, abdominal circumference, and waist-hip ratio (waist circumference/hip circumference) [30]. The secondary outcome was the change from baseline in total cholesterol (TC).

Body weight was measured with a digital scale (OMRON, HBF-514C-LA), while participants wore minimal clothes and no shoes. BMI was calculated as body mass (kg) divided by height in meters squared (kg/m^2^). Waist, hip, and abdominal circumferences (cm) were measured according to standard procedures [31] by using an anthropometric tape (Lufkin W606PM) and were assessed by the same individual to reduce measurement variation.

### 2.7. Statistical Analysis

Because some of the variables had a nonnormal distribution (determined using the Shapiro–Wilk test), statistical analyses were performed with nonparametric tests. Data were expressed as interquartile means, showing the first and third quartiles (percentile 25 to 75). The change from baseline was used to determine the absolute difference between time points and was calculated as the score after the intervention minus the score at baseline for each variable in each volunteer [32]. The categorical values were compared using Fisher’s exact test. The Wilcoxon signed-rank test was used to compare two related samples (intragroup comparisons, before and after analysis). The Mann–Whitney U test was used for between-group comparisons (comparisons of commercial and Wagyu-Cross changes). We used both tests to acquire two types of results for each group; before and after intervention and for an intergroup comparison.

## 3. Results

### 3.1. Food Intervention

Two (6.89%) participants withdrew from the study: one withdrew due to health problems caused by a car accident, and the other was out of town, hindering the collection of post-intervention data. Both withdrawals were from the treatment group (Figure 1). Thus, a total of 27 volunteers completed the food intervention, 14 in the commercial beef group and 13 in the Wagyu-Cross beef group. The analysis included volunteers with BMI ≥ 25 kg/m^2^, thus 20 volunteers: 8 volunteers in the commercial group (4 males and 4 females) and 12 volunteers in the Wagyu-Cross beef group (4 males and 8 females) (*p* = 0.648) were compared.

According to the baseline data, the commercial beef and Wagyu-Cross beef groups did not significantly differ in sex, age, body weight, BMI, waist circumference, hip circumference, abdominal circumference, WHR, or lipid parameters (Table 1).

Since caloric restriction was generated according to their baseline body weight, we stratified the kcal and nutrient intakes according to their BMI for a better comparison. As expected, the dietary records indicated that the main difference in FA intake between the two groups was due to the type of beef consumed, as shown in Table 2. No significant differences between the groups were observed in the dietary FA composition of beef-fat-free intakes (NO BEEF-FAT) included in the menus; in contrast, when comparing the total FA consumption (beef included), significant differences were observed in MUFA and SFA consumption between the groups (Table 2).

### 3.2. Comparison of Anthropometric Characteristics between Groups

Both the Wagyu-Cross beef and commercial beef groups exhibited a decrease in all of the anthropometric characteristics measured with an intragroup significance in all of them, with the exception of the WHR and abdominal circumference in the commercial group, although without significant difference between the groups (Table 3).

### 3.3. Comparison of Lipid Profiles between Groups

At the end of the food intervention, both groups were found to have improved on most of the analyzed lipid parameters (Table 4). The volunteers who consumed Wagyu-Cross beef exhibited a non-significant decrease in serum total cholesterol (Table 4). Additionally, the commercial group exhibited increases in serum low-density lipoprotein cholesterol (LDL-C; 12.5 mg/dL) and non-high-density cholesterol (non-HDL-C; 13.0 mg/dL) and exhibited a significant intragroup increase in serum high-density cholesterol (HDL-C; 6.95 mg/dL) compared to the Wagyu-Cross group.

Even when there were reductions in LDL-C, triglycerides, and non-HDL-C by the Wagyu-Cross volunteers, the decreases were not significant enough to generate significant decreases between the groups in the different atherogenic indices evaluated, although with a significant intragroup decrease in the TC/HDL and Non-HDL/HDL index (Table 4).

In addition to the lipid parameters, a significant difference in the glucose concentration was observed between the groups. Both groups present an increase in glucose concentration after the food intervention. The commercial group exhibited an increase of 11.0 mg/dL compared to the Wagyu-Cross group, which exhibited a non-significant increased glucose concentrations after the food intervention (Table 4). The volunteers’ increased glucose concentrations in the commercial group were significant.

## 4. Discussion

Losing weight is an important goal for people who are obese or overweight, and this change improves health, reducing the risk of most of the obesity-associated complications [33,34]. The effect of beef consumption on weight gain is still controversial. Mixed results have been reported [12,35], but almost as a rule between health professionals, the consumption of red meat is considered as incompatible with a healthy, balanced, weight-loss diet. Our results are interesting because even with the high frequency of beef consumption (four times per week), both groups showed a decrease in weight, similar to that expected in individualized weight-loss therapy, which aims to reduce body weight at an optimal rate of 0.5 to 1 kg per week [36]. Interestingly, substantial weight loss was observed in the Wagyu-Cross beef group (−3.75 kg), and further studies focused on comparing the effects of consumption of beef with a favorable fatty acid composition versus no beef eaters or even with other meat sources will provide better support for our present results.

Similarly, decreases in BMI were also higher in the Wagyu-Cross beef group. BMI and weight reduction are obvious results of caloric restriction; however, studies have claimed that diets including SFAs are more obesogenic than those including unsaturated fatty acids (MUFAs and PUFAs). Moreover, for weight maintenance, a high MUFA or PUFA diet is recommended [37]. Here, we compare the effect of diets differing by up to 16.4% in the fat content provided mainly by the two tested beef types, with the Wagyu-Cross having the highest fat content. We observed a non-significant greater decrease (Table 3) in BMI in the Wagyu-Cross beef group than in the commercial beef group. MUFAs have been reported to be less obesogenic than SFAs since greater fatty acid oxidation is present when the MUFAs are consumed [37]. However, other studies have suggested that FA composition does not have a clear impact on BMI [38,39] or even obesity [40]. The differences observed in our study are important because during the food intervention, all of the volunteers experienced caloric restriction, and the beef FA composition was an important difference between the groups (+8.26% more MUFAs in the Wagyu-Cross beef). These results indicate that frequent beef consumption in conventional diet programs does not have to affect the weight-loss process since we observed weight reductions in both of the groups as long as caloric restriction was maintained.

Waist circumference reduction was higher in the commercial group (Table 3), and a 1-cm increase in WC is associated with a 2% increase in the risk of cardiovascular disease (CVD) [41]. We also observed a significant intragroup decrease in the WHR by the Wagyu-Cross volunteers. High MUFA regimens are known to have a more pronounced impact on fat mass [42]. A review revealed that a Mediterranean diet intervention (characterized by a high intake of PUFAs and, especially, MUFAs as well as a low intake of SFAs) significantly reduces measures of abdominal obesity as determined by waist circumference, the WHR, or visceral fat [43]. Future studies are needed to analyze the effects of higher MUFA consumption by a beef source incorporated into weight-loss programs as a strategy for reducing waist circumference and the WHR, thus reducing CVD risk.

In addition to the effects on anthropometric parameters, replacing dietary SFAs with MUFAs benefits lipid metabolism by reducing total cholesterol, LDL-C, and triglyceride concentrations [24,44]. The consumption of beef high in MUFAs was previously associated with a reduction in cholesterol concentrations [24]. Here, again, the Wagyu-Cross volunteers exhibited a decrease in this parameter although with nonsignificant changes (Table 4); it has been reported that weight loss generates a favorable change in lipid profile by reducing total cholesterol concentrations [45]. We observed an increase in HDL-C concentrations after the food intervention in both of the groups without a significant difference between the groups, although the commercial volunteers presented a higher increase of 6.95 mg/dL in comparison with Wagyu-Cross volunteers (Table 4). Oleic acid consumption has been reported to increase HDL-C concentrations [46,47], and environmental factors, such as diets rich in SFAs and MUFAs as well as weight loss in obese people, can increase HDL-C concentrations [48]. In our study, both of the groups met these conditions; therefore, the inclusion of high MUFA beef in weight-loss diets can be considered a good option given the benefits that the FAs in beef provide for lipid metabolism without compromising the weight loss induced by caloric restriction. Favorable changes in lipid metabolism are an important aspect of caloric restriction, since significant reductions from baseline in LDL-C and triglycerides (TGs) as well as an increase in HDL-C after 6 months (all non-significant) were reported in healthy non-obese individuals compared to controls after caloric restriction [49]. These effects were also observed in our study, especially after Wagyu-Cross beef consumption, so the exclusion or limitation of beef in weight-loss programs should not be mandatory for effective weight loss as long as the caloric restriction is met and, more importantly, to reduce cardiovascular risk as is promoted by the Mediterranean diet to obtain these benefits [6].

Programs focused on weight loss are important due to the health risks of obesity or being overweight, such as the risks of type 2 diabetes mellitus, high blood pressure, and cardiovascular diseases [3]. A key factor in the effectiveness of decreasing body weight is including protein sources due to the greater feelings of satiety and reduced appetite that they produce [18]. Beef is a nutrient-dense and high-quality protein source that plays an important role in helping people meet their essential nutrient needs; thus, the FA composition of meat could be a key point supporting its inclusion in weight-loss diet programs [50]. Here, we observe that after frequent beef consumption in a weight-loss program, volunteers from both of the groups presented a decrease in WC and HC measurements, showing us that beef can be included in these programs and still be effective for weight loss without the need for restriction but more importantly helping to clarify doubts regarding its consumption and the development of an increased risk of several major chronic diseases, such as diabetes and coronary heart disease [51]. We also observed a decrease in triglyceride and VLDL-C concentrations and an increase in HDL-C concentrations (Table 4) after the consumption of both types of beef. The HDL-C increases are a relevant result, since HDL-C can contribute to a cardioprotective effect by removing cholesterol from the peripheral cells [52]. This beneficial effect can be observed in dieters since weight loss significantly increases HDL-C [53] and it has been reported that beef high in MUFAs can increase HDL-C concentration [47], with both good reasons not to exclude beef from weight-loss programs and mainly from a regular diet.

It is well known that high levels of triglycerides and low levels of HDL-C contribute to the development of atherosclerosis [54], a 5–10% weight reduction, and regular exercise programs can effectively reduce triglyceride levels when caloric restriction is accomplished [55]. Our volunteers lost between them about 5% of their weight after the intervention, emphasizing the importance of maintaining a healthy weight and reducing the risk of developing cardiovascular diseases.

Previously, we observed a significant reduction in three atherogenic indices (TC/HDL, LDL/HDL, and non-HDL/HDL) after consumption of Wagyu-Cross beef [24]. Here, we observe that after the consumption of both types of beef, atherogenic indices decrease (Table 4). The relationship between beef consumption and CVD risk is a controversial topic and is still under investigation. The inconsistent findings on these associations are partly explained by different factors, including the measuring and cooking methods and the null differentiation between beef, pork, and lamb meat, taking it in general as red meat consumption [16,56,57,58]. The cooking methods and temperatures are important factors to consider since the practice of cooking meat at high temperatures (e.g., pan frying, grilling, and barbecuing) may lead to the production of heterocyclic amines (HAAs), which are thought to increase colon cancer risk [59,60], but the generation of HAAs is not exclusive to red meat, since they are also produced during the frying of fish or chicken [51].

An interesting result was the significant intragroup changes in glucose concentration. In the commercial beef group, an increase of 11.50 mg/dL was observed (Table 4), while the volunteers of Wagyu-Cross beef had an increase of 0.50 mg/dL, thus a significant difference of 11.0 mg/dL between the groups was obtained. There is emerging evidence that meat and processed meat consumption could be associated with the development of type 2 diabetes mellitus (T2DM) [51]. Although the exact mechanism is still unclear, a hypothesis states that beef SFA and cholesterol could increase insulin resistance [61]; thus, beta-cells need to produce a significant amount of insulin for glucose disposal [62]. These implications have been observed in two controlled intervention studies, including healthy volunteers and volunteers with insulin resistance. After the consumption of an SFA-rich diet or a MUFA-rich diet, the consumption of the SFA-rich diet impaired insulin sensitivity [63,64]. Nevertheless, when the duration of the diet intervention increased, no significant change was observed [65]. Further studies are needed to determine whether SFA from beef contributes to insulin resistance and whether the minor effect observed in the Wagyu-Cross group is also related to its MUFA content (particularly oleic acid). The results of this study should be taken with caution due to the limited sample size and the fact that lifestyle recommendations remain the first-line therapy.

Finally, in addition to health-related concerns, as a production system, beef is facing enormous challenges due to its negative association with environmental effects such as greenhouse gas emissions and biodiversity loss [66,67]. These problems are not disregarded by cattle farmers, who, in concordance with animal science, are promoting production strategies that contribute to diminishing the negative aspects of its production [68,69]. Meanwhile, it is necessary to continue studies focused on determining the impact of beef on consumer health, especially if we consider that global beef consumption has increased [70], showing a greater preference for meat and the enjoyment of it [71]. Our study is focused on providing support for beef consumption as a millenary source of natural protein. We support the idea that its adequate consumption in quantity and quality will be beneficial to consumers who prefer to include beef in their diets.

## 5. Conclusions

The consumption of beef even four times a week did not compromise the volunteers’ weight loss. The beef’s FA composition had no significant difference in the anthropometric measures between the groups, but higher weight loss and decreases in BMI were observed in the Wagyu-Cross beef group than in the commercial beef group (0.85 kg and 0.46 kg/m², respectively).

High MUFA beef is proposed as a favorable source of protein showing no significant repercussions on the lipid metabolism of consumers when a weight-loss program with specific nutritional indications was followed.

## Figures and Tables

**Figure 1 nutrients-14-03711-f001:**
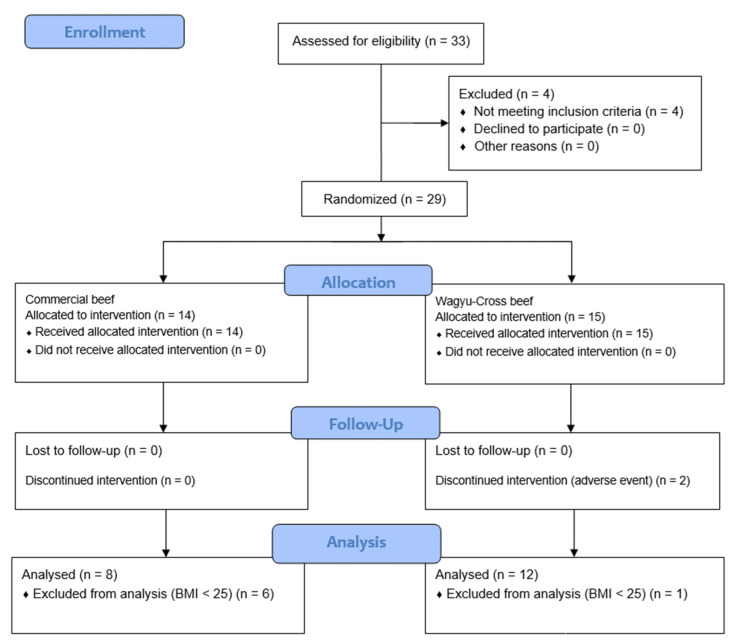
CONSORT flow diagram of the numbers of volunteers screened, included, eliminated, and analyzed in the present study.

**Table 1 nutrients-14-03711-t001:** Baseline anthropometric and biochemical characteristics of the volunteers ^1^.

Clinic Characteristic	Group		*p* Value *
	Commercial Beef (SD)	Wagyu-Cross Beef (SD)	
Age, Years	34.5 (27.25–42.50)	42.0 (32.50–49.50)	0.098
Body weight, kg	81.80 (72.60–100.18)	75.15 (71.73–94.80)	0.571
BMI (kg/m^2^)	30.41 (25.50–32.73)	30.07 (26.31–34.29)	1.000
Waist circumference, cm	95.00 (85.88–105.38)	90.00 (81.50–100.80)	0.473
Hip circumference, cm	112.45 (99.38–121.73)	108.85 (102.63–110.75)	0.571
WHR	0.85 (0.79–0.88)	0.87 (0.75–0.92)	0.910
Abdominal circumference, cm	108.25 (91.63–112.00)	96.75 (93.25–110.78)	0.678
Cholesterol (mg/dL)	169.0 (155.75–194.75)	170.50 (150.50–197.0)	1.000
Triglycerides (mg/dL)	136.50 (99.75–196.00)	123.0 (88.75–170.48)	0.678
HDL-C (mg/dL)	38.50 (35.50–46.00)	40.00 (35.83–47.75)	0.571
LDL-C (mg/dL)	105.50 (94.00–114.75)	100.50 (84.50–124.25)	0.624
VLDL-C (mg/dL)	27.00 (19.75–39.00)	24.50 (18.00–30.45)	0.678
Non-HDL-C	129.5 (120.00–150.00)	128.50 (112.0–153.0)	0.851
TC/HDL-C	4.41 (3.77–4.86)	3.98 (3.71–4.82)	0.571
LDL-C/HDL-C	2.88 (2.10–2.95)	2.52 (2.05–3.16)	0.678
Non-HDL-C/HDL-C	3.41 (2.77–3.86)	2.98 (2.71–3.82)	0.571
TC * TG * LDL-C/HDL-C	16.20 (10.87–27.77)	16.07 (10.19–24.06)	0.851
Glucose (mg/dL)	99.00 (89.25–101.75)	97.5 (89.88–106.50)	0.678

Abbreviations: BMI: body mass index; WHR: waist-to-hip ratio; TC: total cholesterol; HDL-C: high-density lipoprotein cholesterol; LDL-C: low-density lipoprotein cholesterol; VLDL-C: very low-density lipoprotein cholesterol; TG: triglycerides. ^1^ Values shown are interquartile means and the first and third quartiles. * U-Mann–Whitney test.

**Table 2 nutrients-14-03711-t002:** Volunteers’ intake of fat and FA composition during the intervention ^1^.

BMI	Macronutrient	Group		*p* Value *
		Commercial Beef	Wagyu-Cross Beef	
≥25–≤29.9 kg/m^2^	Kcal	1297.17 (88.30)	1246.55 (49.27)	0.293
	Lipids, %	29.69 (1.57)	35.26 (1.86)	0.003
	MUFA, g	15.17 (2.35)	20.03 (1.10)	0.003
	SFA, g	12.06 (0.36)	16.44 (0.53)	0.000
	PUFA, g	4.45 (0.82)	4.44 (0.14)	0.984
	MUFA, g (NO BEEF-FAT)	8.70 (2.78)	6.52 (2.24)	0.240
	SFA, g (NO BEEF-FAT)	5.71 (1.66)	4.83 (1.11)	0.366
	PUFA, g (NO BEEF-FAT)	3.89 (1.13)	3.48 (0.33)	0.414
≥30 kg/m^2^	Kcal	1535.13 (149.85)	1424.11 (193.06)	0.323
	Lipids, %	28.60 (1.53)	34.09 (1.44)	0.000
	MUFA, g	16.30 (1.88)	21.45 (1.56)	0.001
	SFA, g	13.37 (1.31)	17.82 (1.21)	0.000
	PUFA, g	5.26 (1.03)	4.91 (0.60)	0.497
	MUFA, g (NO BEEF-FAT)	8.50 (1.88)	6.33 (1.56)	0.541
	SFA, g (NO BEEF-FAT)	6.25 (1.31)	5.19 (1.21)	0.863
	PUFA, g (NO BEEF-FAT)	4.48 (1.03)	3.43 (0.60)	0.278

^1^ Values presented are means and standard deviations. * Student’s *t*-test. Abbreviations: FA: fatty acids; BMI: body mass index; Kcal: kilocalorie; MUFA: monounsaturated fatty acid; SFA: saturated fatty acid; PUFA: polyunsaturated fatty acid; g: grams; NO BEEF-FAT: beef-fat-free intakes.

**Table 3 nutrients-14-03711-t003:** Comparison of anthropometric measurements before and after food intervention ^1^.

Clinical Characteristic	Group		*p* Value ^†^
	Commercial Beef	Wagyu-Cross Beef	
Weight (kg)			
Baseline	81.80 (72.60–100.18)	75.15 (71.73–94.80)	
Final	77.60 (69.73–96.68)	71.45 (67.93–92.55)	
*p* Value ^	0.036	0.002	
Change	−2.90 (−4.25 to −0.93)	−3.75 (−5.38 to −2.95)	0.297
BMI (kg/m^2^)			
Baseline	30.41 (25.50–32.73)	30.07 (26.31–34.29)	
Final	28.76 (26.00–31.17)	28.34 (24.78–33.01)	
*p* Value ^	0.036	0.002	
Change	−1.03 (−1.66 to −0.30)	−1.49 (−2.05 to −1.06)	0.203
WC *			
Baseline	95.00 (85.88–105.38)	90.00 (81.50–100.80)	
Final	87.00 (79.10–98.50)	85.00 (74.38–96.48)	
*p* Value ^	0.012	0.002	
Change	−6.75 (−7.80 to −4.75)	−4.80 (−9.25 to −3.00)	0.246
HC *			
Baseline	112.45 (99.38–121.73)	108.85 (102.63–110.75)	
Final	106.00 (87.38–118.05)	104.65 (100.05–108.68)	
*p* Value ^	0.012	0.005	
Change	−6.45 (−15.15 to −2.55)	−4.10 (−4.88 to −2.00)	0.189
WHR			
Baseline	0.85 (0.79–0.88)	0.87 (0.75–0.92)	
Final	0.85 (0.81–0.90)	0.84 (0.74–0.89)	
*p* Value ^	0.674	0.014	
Change	−0.02 (−0.04 to 0.07)	−0.02 (−0.06 to −0.01)	0.430
AC *			
Baseline	108.25 (91.63–112.00)	96.75 (93.25–110.78)	
Final	100.20 (98.23–109.43)	91.15 (87.25–106.00)	
*p* Value ^	0.484	0.004	
Change	−3.05 (−7.33 to 5.45)	−5.25 (−6.75 to −3.68)	0.203

Abbreviations: BMI: body mass index; WC: waist circumference, HC: hip circumference; WHR: waist-to-hip ratio; AC: abdominal circumference. ^1^ All values are interquartile means and the first and third quartiles. * cm. ^ *p* Value: Baseline versus final, Wilcoxon test (two related samples). ^†^ *p* Value: commercial versus Wagyu-Cross, U-Mann–Whitney test (independent samples).

**Table 4 nutrients-14-03711-t004:** Comparison of serum lipid parameters and atherogenic indices before and after food intervention ^1^.

Clinical Characteristic	Group		*p* Value ^†^
	Commercial Beef	Wagyu-Cross Beef	
Cholesterol *			
Baseline	169.0 (155.75–194.75)	170.50 (150.50–197.0)	
Final	174.00 (144.0–182.50)	161.5 (143.75–186.65)	
*p* Value ^	0.612	0.209	
Change	4.00 (−34.00 to 23.75)	−11.00 (−18.60 to 8.75)	0.700
HDL-C *			
Baseline	38.50 (35.50–46.00)	40.00 (35.83–47.75)	
Final	48.50 (40.75–54.00)	41.00 (33.80–46.75)	
*p* Value ^	0.021	0.556	
Change	8.50 (4.50 to 11.75)	1.55 (−4.75 to 8.00)	0.069
LDL-C *			
Baseline	105.50 (94.00–114.75)	100.50 (84.50–124.25)	
Final	101.50 (78.25–114.75)	90.50 (82.25–122.78)	
*p* Value ^	0.575	0.666	
Change	4.50 (−44.00 to 20.75)	−8.00 (−14.00 to 11.00)	0.877
VLDL-C *			
Baseline	27.00 (19.75–39.00)	24.50 (18.00–30.45)	
Final	20.00 (14.50–29.25)	20.00 (16.50–25.50)	
*p* Value ^	0.034	0.025	
Change	−5.50 (−8.75 to −0.75)	−4.91 (−7.50 to 1.25)	0.698
TG *			
Baseline	136.50 (99.75–196.00)	123.0 (88.75–170.48)	
Final	103.00 (73.00–149.50)	101.50 (82.00–150.85)	
*p* Value ^	0.025	0.023	
Change	−26.00 (−42.25 to −5.50)	−21.75 (−38.00 to 2.50)	0.616
Non-HDL-C			
Baseline	129.5 (120.00–150.00)	128.50 (112.0–153.0)	
Final	131.50 (95.25–135.00)	124.0 (104.75–144.25)	
*p* Value ^	0.575	0.158	
Change	2.00 (−44.75 to 15.00)	−11.00 (−18.68 to 5.50)	0.847
TC/HDL-C			
Baseline	4.41 (3.77–4.86)	3.98 (3.71–4.82)	
Final	3.75 (2.91–4.36)	3.83 (3.57–4.42)	
*p* Value ^	0.069	0.041	
Change	−0.55 (−1.64 to 0.01)	−0.38 (−0.63 to −0.11)	0.335
LDL-C/HDL-C			
Baseline	2.88 (2.10–2.95)	2.52 (2.05–3.16)	
Final	2.20 (1.60–2.69)	2.36 (2.08–2.75)	
*p* Value ^	0.208	0.084	
Change	−0.39 (−1.54 to 0.28)	−0.25 (−0.50 to 0.03)	0.512
Non-HDL-C/HDL-C			
Baseline	3.41 (2.77–3.86)	2.98 (2.71–3.82)	
Final	2.75 (1.91–3.36)	2.83 (2.57–3.42)	
*p* Value ^	0.069	0.041	
Change	−0.55 (−1.64 to 0.01)	−0.38 (−0.63 to −0.11)	0.335
TC * TG * LDL-C/HDL-C			
Baseline	16.20 (10.87–27.77)	16.07 (10.19–24.06)	
Final	12.88 (4.74–22.25)	11.48 (9.62–16.75)	
*p* Value ^	0.050	0.084	
Change	−2.77 (−9.19 to 0.35)	−4.42 (−7.50 to 1.15)	0.939
Glucose *			
Baseline	99.00 (89.25–101.75)	97.5 (89.88–106.50)	
Final	103.50 (97.50–116.00)	94.95 (82.50–109.00)	
*p* Value ^	0.036	0.480	
Change	11.50 (0.25 to 14.50)	0.50 (−15.75 to 7.10)	0.013

Abbreviations: TC: total cholesterol; HDL-C: high-density lipoprotein cholesterol; LDL-C: low-density lipoprotein cholesterol; VLDL-C: very low-density lipoprotein cholesterol; TG: triglycerides. ^1^ All values are interquartile means and the first and third quartiles. * cm. ^ *p* Value: Baseline versus final, Wilcoxon test (two related samples). ^†^ *p* Value: commercial versus Wagyu-Cross, U-Mann–Whitney test (independent samples).

## Data Availability

Not applicable.

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
