# Peer review of "Effect of Consuming Beef with Varying Fatty Acid Compositions as a Major Source of Protein in Volunteers under a Personalized Nutritional Program"

_nutrients, 2022, doi:10.3390/nu14183711_

Round 1

Reviewer 1 Report (Previous Reviewer 1)

The authors have addressed all of my comments and the paper is ready for publication. 

Author Response

Reviewer 1

The authors have addressed all of my comments and the paper is ready for publication. 

Response from authors:

We appreciate your comments on our manuscript

Reviewer 2 Report (New Reviewer)

The authors investigated the effect of consuming beef with two different fat- and MUFA contents on anthropometric and biochemical values in a personalised diet. It is an interesting study, however, several question/concerns were raised.

1.) In rows 73-75 you write about high MUFA content of the Wagyu-Cross beef compared to the commercial beef, but what is the MUFA (or at least oleic acid) content of the commercial beef? And similarly: you write about the high SFA content of commercial beef, but what is the SFA content of the Wagyu-Cross beef? These values should be compared pairwise. By the SFA you mention palmitic acid (C16:0) and C15:0, that is a minor fraction. Why not stearic acid that is the second highest value? These two FAs (C16:0 and C18:0) are responsible for most of the SFAs. I see from your previously published article that these two (C15:0 and C16:0) were only significantly different between the two beef samples but you should focus here on the main FAs.

2.) Based on the data from your previous publication, the difference between the fatty acid composition of the two beefs is not really significant (from the physiological point of view). The commercial meat and the Wagyu-Cross beef has 45.2% and 43.0% SFA as well as 49.6% and 51.4% MUFA respectively. The p values for SFA and MUFA are quite low (0.021 and 0.036). If you also take into account the fact that conventional beef is lower in fat (13.1 g/100 g), then in principle Wagyu-Cross beef (24.5 g/100 g) should provide much more palmitic and stearic acid (10.5 g SFA / 100 g beef) than conventional beef (5.9 g SFA / 100 g beef). How can you explain this and say that eating Wagyu-Cross beef is healthier and more beneficial?

3.) The BMI of the potentially appropriate, included person are too wide (between 18.5-45 kg/m2), including person with normal BMI, overweight, obese and morbid obese. How many from each (BMI) group were selected to the two different beef-diet groups respectively? Were the proportions of the (BMI) subgroups in the two diet groups comparable? Can this heterogeneity of the included person not lead to biases?

4.) There is no information about the fatty acid composition of the diet. Not only beef, but other components of the diet can be rich in SFAs and/or MUFAs, like oils, nuts, vegetables, other meats. As you gave the diet plan to the participants and made a weekly dietary monitoring (as stated in the article) you should be able to make a table of the main FA composition of the diet in the two investigation groups (SFA, MUFA, n-3 and n-6 PUFAs, C16:0, C18:0, C18:1n-9, C18:2n-6, C20:4n-6, C18:3n-3, C20:5n-3, C22:6n-3). Were there any differences between the two groups in the overall fatty acid intake? Because the difference between the FA composition of the two beef types may not be equal to the difference between the combined fatty acid intakes of the two dietary groups (rows 269-272). Please clarify it with an average weekly FA intake of the two dietary groups (as a Table if possible). Without these data I feel your discussion and conclusion about the effect of FA difference in the diets rather hypothetic.

Some minor comments:

- In Table 1 what does it mean that sex of participants was 0.57 vs.0.62? How many men and women participated in each group?

- You found that there was an increase in cholesterol in the commercial beef group and a decrease in the Wagyu-Cross beef group, however these changes are really small (from 178.93 [39.45] to 182.57 [32.16] and from 169.57 [29.88] to 164.35 [26.37]). It is more informative than the mean (SD) in this case to observe the direction of change for all participants. In how many cases did cholesterol increase in the conventional beef diet group (in how many out of 14 people) and how many cases did it decrease in the Wagyu-Cross diet group (in how many out of 13 people). In my opinion a few large changes (in one or two people) could mask the real direction of change (can it be a biased result?).

Author Response

Reviewer 3 Report (New Reviewer)

This randomized controlled clinical trial evaluated anthropometric and serum lipid changes after a food intervention that included frequent beef consumption for four days per week for four weeks. Volunteers were randomly assigned to the commercial or Wagyu-Cross beef groups, with the latter beef possessing higher fat and MUFA contents.

In spite of the limited volunteers included, the study was well designed and had a good dietary assessment. The results and outcome measurements of the anthropometric, blood lipids and glucose gave an overview how different fatty acid saturation levels can influence these outcomes.

The suggestion that beef should be included in weight loss programs since it facilitate weight loss without negatively influencing on lipid metabolism is correct But, if the glucose level increases within four weeks, losing weight may not be beneficial for human health.

In the last years, many studies showed that gut microbiota have a pig effect on human health and glucose homeostasis. The specific short chain fatty acids (SCFA) profile produced by microbial fermentation shall be used as one attribute to characterize the nutritional properties of a special diet. This aspect in relationship with a red meat diet was not discussed in this study.

I agree with the authors recommendation, that further studies focused to compare the effects of consumption of beef with a favourable fatty acid composition versus no beef eaters or even with other meat sources will give a better support to the present results.

Author Response

Please see the attachment."

Round 2

Reviewer 2 Report (New Reviewer)

The authors revised and recalculated the results and answered all my questions. Now the article is more convincing and Table 2 about dietary intake with and without beef proves that the only difference in the fatty acid intake arise from the different fatty acid composition of commercial and Wagyu-Cross beef. It is an interesting study, only some minor comments were raised:

·         Row 39-40: you state that the link between dietary pattern and diabetes have controversial results, but without any reference. The next sentence (or study) in not about diabetic person. As you didn’t specify your study to diabetic, overweight adults, do you really need this sentence? If yes, please give some references with positive, negative and/or no correlation between diet and diabetes. If not, please delete it.

·         Row 40-42 (Ref 7): this sentence is for me misleading: the diet resulted in smaller increase in BMI and WC. Smaller increase than what? The authors investigated 6 different dietary patterns, so did you think that than the other 5? Or what? An is it a positive result? Is it good for us?

·         Ref 11: webpage cannot be found. Please either give a new link or give another proper reference (maybe an article published in a Pubmed/Scopus/Google scholar indexed journal or a guideline that can be downloaded).

·         Ref 12: link or at least ISBN/journal with pages where this publication can be found. Please cite it properly!

·         Row 57-59: the cited meta-analysis is about processed and unprocessed red meat consumption and not only beef. Please correct it.

·         Row 264 (and row 371): where comes ‘decrease of 12.43 mg/dL (glucose)’ in the Wagyu-Cross group from (baseline mean was 99.29 mg/dL and final 96.24 mg/dL (change -3.05))? Please clarify it or reword the sentence! It is a difference between which two values?

Author Response

This manuscript is a resubmission of an earlier submission. The following is a list of the peer review reports and author responses from that submission.

Round 1

Reviewer 1 Report

Comments to the authors:

1. The introduction does not present all the crucial information regarding beef consumption and disease risk. Accumulating evidence from large scale population studies shows that diet rich in red meat greatly increase the risk for heart disease and colorectal cancer. This information should be discussed in detail in the introduction and discussion.

https://www.nih.gov/news-events/nih-research-matters/eating-red-meat-daily-triples-heart-disease-related-chemical

https://www.cancer.gov/news-events/cancer-currents-blog/2021/red-meat-colorectal-cancer-genetic-signature

2. Moreover, regular consumption of red meat has devasting effects on the climate, and is one of the main drivers of climate change. Livestock production does not only have a negative influence on green house gas emissions, but also on the water footprint, water pollution, and water scarcity. This should be discussed in the introduction and conclusion.

https://www.ncbi.nlm.nih.gov/pmc/articles/PMC7256495/

3. The study is very limited in that it does not have a control group that does not consume red meat. It would be interesting to see how lipid profiles change in the red meat eaters, versus controls.

4. No power calculation is presented. How was the sample size determined?

Minor comments:

5. Line 40: The sentence is a bit difficult to understand: “On the other hand, changes are also observed in individuals 40 following a dietary pattern with low-fat foods (red meat, margarine, butter, etc.). Butter etc is obviously not a low fat food, please revise.

Reviewer 2 Report

Vela-Vásquez et al in their present study titled “Effect of consuming beef with varying fatty acid compositions 2 as a major source of protein on weight loss volunteers” have aimed to study the effects of inclusion of beef should in weight loss programs as it may facilitate weight loss without negatively influencing lipid metabolism. The authors in their study have given a good introduction of the concept they would like to introduce and overall it was interesting to read the manuscript. Overall the manuscript is well written and statistical analysis is adequate. A few minor corrections are needed

Introduction:

The authors have provided a single reference for the inclusion of beef in their diet. Is this the only study and if not, then more references need to be added. A paragraph on the different types of beef and their characteristics is needed to give a general understanding on these classes differ and why it is important to ‘know your meat’.

Methodology:

The study design and methodology are clearly described. Lines 101 is repeated and is same as Line 86 and 87. Please clarify how you are able to mention that consuming beef for “four days per week, provided “29.5% of dietary fat for the commercial group 104 and 34.8% of fat for the Wagyu-Cross group”.

It would be more feasible to mention under the Beef processing section only the details of the meat was obtained, processed and then packed for delivery to the participants needs to be included. The lines 125-132 can be added to the introduction where I mentioned that a brief introduction about the type of meats needs to be added.

The methodology is missing the details of how the biochemical analysis of the lipids was carried out. This needs to be added.

Discussion:

Line 346 should be a separate paragraph. 

Reviewer 3 Report

The paper analyzed the effect of consuming beef with different fatty acid composition, commercial beef vs Wagyu-Croos beef on anthropometric and biochemical parameters, showing that beef consumption (4 times for 4 weeks) facilitated weight loss and increase HDL.

The paper described a poorly conducted study, with scanty analysis the analysis date poorly interpreted. First of all, the sample size of the study group is also too small. In addition, it lacks a group without beef consuming and the same diet of the two group of the study, to understand if the effect of weight loss is strictly dependent of beef consumption. It should be noticed that the authors analyzed first the two group of study independently, but this for most parameters didn’t shown significance, then the analysis was performed in a unique group, probably to improve statistical power. For this reason it lacks a correct control group.

In particular I have the following specific comments:

- Regard the Methods section, the dietary assessment of the study group is not clear. For instance lines 105-106 “7-day diet” is during the 4 week food intervention with beef? The caloric restriction is not described. Lack description of all biochemical data in terms of assay (type of assay, intra-assay and inter-assay coefficients of variation).

- In the results there are major discrepancies within text and the tables. For instance lines 207-210 (ex LDL-C -0.87 in the text but in the table -2.71)

- Regard statistical analysis the two type of t-test are not clear explained.

- The conclusion of the author is purely speculative and not supported by the study data.

- Finally the discussion is too long and in some point is confused and to speculative (Unqualified statements are made that are not supported by the data)

- The literature is out of date.

Round 2

Reviewer 1 Report

The authors have done an excellent job addressing all of my comments.

Reviewer 3 Report

The revision that has been made is not sufficient for accepting the paper. The  main point is that it still lacks a group without beef consuming and the same diet of the two group of the study, to understand if the effect of weight loss is strictly dependent of beef consumption. The biochemical parameters are still not well described   - In the results there are still major discrepancies within text and the tables. The conclusion of the author is purely speculative and not supported by the study data.